# Humic Acids Preparation, Characterization, and Their Potential Adsorption Capacity for Aflatoxin B_1_ in an In Vitro Poultry Digestive Model

**DOI:** 10.3390/toxins15020083

**Published:** 2023-01-17

**Authors:** Jesús Adonai Maguey-González, María de Jesús Nava-Ramírez, Sergio Gómez-Rosales, María de Lourdes Ángeles, Bruno Solís-Cruz, Daniel Hernández-Patlán, Rubén Merino-Guzmán, Xóchitl Hernández-Velasco, Juan de Dios Figueroa-Cárdenas, Alma Vázquez-Durán, Billy M. Hargis, Guillermo Téllez-Isaías, Abraham Méndez-Albores

**Affiliations:** 1Posgrado en Ciencias de la Producción y de la Salud Animal, Universidad Nacional Autónoma de México (UNAM), Unidad de Posgrado, Ciudad Universitaria, Coyoacán, Ciudad de México 04510, Mexico; 2Unidad de Investigación Multidisciplinaria L14 (Alimentos, Micotoxinas, y Micotoxicosis), Facultad de Estudios Superiores Cuautitlán, Universidad Nacional Autónoma de México (UNAM), Cuautitlán Izcalli 54714, Mexico; 3Centro Nacional de Investigación Disciplinaria en Fisiología y Mejoramiento Animal (CENID-INIFAP), Km1 Carretera a Colon Ajuchitlán, Querétaro 76280, Mexico; 4Laboratorio 5: LEDEFAR, Unidad de Investigación Multidisciplinaria, Facultad de Estudios Superiores (FES) Cuautitlán, Universidad Nacional Autónoma de México (UNAM), Cuautitlán Izcalli 54714, Mexico; 5División de Ingeniería en Nanotecnología, Universidad Politécnica del Valle de México, Tultitlan 54910, Mexico; 6Departamento de Medicina y Zootecnia de Aves, Facultad de Medicina Veterinaria y Zootecnia, Universidad Nacional Autónoma de México (UNAM), Ciudad de México 04510, Mexico; 7CINVESTAV-Unidad Querétaro, Libramiento Norponiente No. 2000, Fraccionamiento Real de Juriquilla, Querétaro 76230, Mexico; 8Department of Poultry science, University of Arkansas, Fayetteville, AR 72701, USA

**Keywords:** adsorption, aflatoxin B_1_, humic acids, in vitro digestion model

## Abstract

Vermicompost was used for humic acid (HA) preparation, and the adsorption of aflatoxin B_1_ (AFB_1_) was investigated. Two forms of HA were evaluated, natural HA and sodium-free HA (SFHA). As a reference, a non-commercial zeolitic material was employed. The adsorbents were characterized by attenuated total reflectance-Fourier transform infrared spectroscopy (ATR-FTIR), energy-dispersive X-ray spectroscopy (EDS), zeta potential (ζ-potential), scanning electron microscopy (SEM), and point of zero charge (pHpzc). The adsorbent capacity of the materials when added to an AFB_1_-contaminated diet (100 µg AFB_1_/kg) was evaluated using an in vitro model that simulates the digestive tract of chickens. Characterization results revealed the primary functional groups in HA and SFHA were carboxyl and phenol. Furthermore, adsorbents have a highly negative ζ-potential at the three simulated pH values. Therefore, it appears the main influencing factors for AFB_1_ adsorption are electrostatic interactions and hydrogen bonding. Moreover, the bioavailability of AFB_1_ in the intestinal section was dramatically decreased when sorbents were added to the diet (0.2%, *w*/*w*). The highest AFB_1_ adsorption percentages using HA and SFHA were 97.6% and 99.7%, respectively. The zeolitic material had a considerable adsorption (81.5%). From these results, it can be concluded that HA and SFHA from vermicompost could be used as potential adsorbents to remove AFB_1_ from contaminated feeds.

## 1. Introduction

Some toxigenic fungi can synthesize mycotoxins as secondary metabolites; mycotoxins have a wide range of chemical structures and a low molecular weight and threaten human and animal health [1]. The most common way to consume mycotoxins is through food contamination; however, exposure can also happen when spores are directly contacted or inhaled [2]. Of all the mycotoxins, aflatoxin B_1_ (AFB_1_) is the most harmful. Two closely related fungi, *Aspergillus flavus* and *Aspergillus parasiticus,* produce primarily aflatoxins. For instance, *A. togoensis* only synthesizes AFB_1_; *A. flavus* and *A. pseudotamarii* synthesize AFB_1_ and aflatoxin B_2_ (AFB_2_), while *A. aflatoxiformans*, *A. arachidicola*, *A. austwickii*, *A. cerealis*, *A. luteovirescens*, *A. minisclerotigenes*, *A. mottae*, *A. nomius*, *A. novoparasiticus*, *A. parasiticus*, *A. pipericola*, *A. pseudocaelatus*, *A. pseudonomius*, *A. sergii*, and *A. transmontanensis* produce AFB_1_, AFB_2_, aflatoxin G_1_ (AFG_1_), and aflatoxin G_2_ (AFG_2_) [3]. The toxic effects of AFB_1_ on poultry are well known. Low production and a high vulnerability to illness are hazards of AFB_1_ in poultry [2]. Hepatotoxic consequences include reduced liver-to-body weight ratios, changes in liver enzymes, abnormal blood-clotting pattern, and histological abnormalities, such hepatocellular necrosis and biliary hyperplasia [4].

The detoxification of mycotoxin-contaminated grains can be accomplished using a variety of procedures, including physical removal, chemical conversion to less toxic products, enzymatic detoxification, and microbial degradation, among others [5]. A better approach to reducing the harmful effects of mycotoxins on animal health would be to include more natural active ingredients in the feed. In the study of mycotoxin binders, substances derived from plants play a significant role [5,6]. Additionally, decontamination procedures must be integrated into animal diets in a simple, affordable, and safe manner Physical methods of removing or inactivating mycotoxins are less expensive and easier to use than chemical methods [7]. The effectiveness of adsorbents is related to their structure, charge distribution, and surface area. The shape and polarity of mycotoxins also affect their binding affinity [8].

Humic substances (HS), mainly composed of humic acids (HA), fulvic acids (FA), and humins, are heterogeneous macromolecules with numerous negatively charged functional groups, primarily carboxyl and phenol [9]. These groups are thought to be potential pollutant-binding sites, such as metallic species, herbicides, and pesticides [10], as well as complex formations with metal cations [11]. Recently, our research group studied an extract of HA from vermicompost under different experimental conditions to clarify its mechanism of action [12,13,14]. In previous in vitro and in vivo studies, several sources of HS containing mixtures of HA and FA have been used as mycotoxin binders [15]; however, purified HA from vermicompost has never been tested against AFB_1_. HAs extracted from vermicompost are still in an early humification process and are considered immature; this factor may cause structural differences compared to HA extracted from lignites, leonardites, or other HA-aged sources.

Attenuated total reflectance-Fourier transform infrared spectroscopy (ATR-FTIR), energy-dispersive X-ray spectroscopy (EDS), zeta potential (ζ-potential), scanning electron microscopy (SEM), and point of zero charge (pHpzc) are some important techniques for characterizing mycotoxins binders [16]. These techniques involve little material and are easily repeatable, non-destructive, and reasonably straightforward. These techniques may help to characterize the main chemical components of HA extracted from vermicompost, which may help to elucidate the possible mechanisms of action of HA as potential aflatoxin binder. As a result, the purpose of this study was to prepare and characterize HA from vermicompost and evaluate its adsorption capacity for AFB_1_ in an in vitro poultry digestive model.

## 2. Results and Discussion

### 2.1. Characterization

#### 2.1.1. ATR-FTIR

Figure 1 shows the ATR-FTIR spectra of HA, SFHA, and the zeolite. The main FTIR bands and their corresponding assignments are shown in Table 1. In general, both HA adsorbents have a wide variety of functional groups, and their spectra showed high intensity for five principal bands: (A) 3238–3267 cm^−1^ associated with OH-stretching vibrations, (B) 2926–2927 cm^−1^ related to aliphatic groups, (C) 1571–1957 cm^−1^ associated with carboxyl, amide, and aromatic vibrations, (E) 1412–1420 cm^−1^ associated with carboxyl, aromatic, and phenol vibrations, and (G) 1120–1123 cm^−1^ associated with C-O carbonyl. Furthermore, HA shows two distinctive bands at (I) 846 cm^−1^ and (J) 620 cm^−1^ assigned to the aromatic and aliphatic groups. The lack of certain bands in the FTIR spectra indicates a lower content of functional groups. Therefore, the absence of some bands in the SFHA spectra indicates HA has higher quantities of aromatic and aliphatic groups. On the other hand, SFHA exhibits three specific bands at (D) 1508 cm^−1^ (carboxyl, amide, and aromatic groups), (F) 1215 cm^−1^ (carboxyl vibrations), and (H) 1030 cm^−1^ (aromatic groups). The band located between 1100 cm^−1^ and 1035 cm^−1^ corresponds to the Si-O stretching of silicate [17]. Its presence is attributed to impurities of aluminosilicates not eliminated during the HA extraction process from soils. However, the non-commercial zeolitic material also exhibited the distinctive band associated with Al^3+^ -OH at (A) 3621 cm^−1^. Significant water absorption at the B (3387 cm^−1^) and C bands (1630 cm^−1^) proved the zeolite was also hydrated. These bands (B and C) are commonly associated with water molecules linked with Na^+^ and Ca^2+^ in the channels and cages of the zeolitic material. The band at (D) 1000 cm^−1^ corresponds to the stretching vibration of T–O in TO^4^ tetrahedra (T = Si and Al). Furthermore, the bands at (E) 793 cm^−1^ and (H) 444 cm^−1^ are related to the stretching vibration of O–T–O and the bending of T–O bonds, respectively [17]. Finally, the absorption band at (F) 600 cm^−1^ is associated with the presence of heulandite [18]. These results are in accordance with those obtained by other researchers [19,20,21,22,23]. In general, HA contains a wide variety of acidic functional groups, such as carboxylic, carbonyl, hydroxyl, and phenolic (hydrophilic domains), as well as methyl, aliphatic, and aromatic moieties (hydrophobic domains). These functional groups are considered potential sites for binding pollutants [11,24], and for building complexes with certain metal cations [10]. Other studies indicate OH, C-O, C=O, and phenolic groups form hydrogen bonds with certain pollutants [17,25,26]. For instance, Vázquez-Durán et al [27] indicated OH could establish hydrogen bonds with the oxygen atoms in the methoxy, carbonyl, and ether groups of the AFB_1_ molecule.

#### 2.1.2. SEM

The surface morphology and microstructure of the sorbents were evaluated using a series of acquired SEM images (Figure 2). The surface morphology of the zeolite is amorphous, and large clumps cover the pores. On the other hand, HA and SFHA had rough and uneven surfaces with aggregates of different shapes and sizes. Specifically for HA, most of the aggregated particles were approximately 163.42 ± 20.14 μm in size. Moreover, grooves on the particle surface with the trace of granules were also observed. These results are consistent with the granular structure of several HAs extracted from various sources [28,29]. HA’s physical and chemical compositions differ considerably depending on its source, environmental conditions, and extraction procedure [20,24]. Several variables, including concentration, pH, ionic strength, charge density, acidic group ionization degree, and intermolecular interactions, influence the structural conformation of HA [30,31]. The HA structure is protonated at low pHs (below 4), resulting in a more condensed structure due to the establishment of H-bonds and van der Waals interactions. However, at pH levels between 4 and 7, HA adopts a more expanded and scattered macrostructure due to intra and intermolecular repulsion. Furthermore, at pH greater than 11, a condensed structure can also be observed [32]. According to this research, certain HA molecules engage in supramolecular interactions through dispersive forces, such as hydrogen bonds, van der Waals interactions, and π–π interactions [28,33].

#### 2.1.3. EDS

Figure 3 shows the energy-dispersive X-ray spectroscopy (EDS) spectra of the HA, SFHA, and the zeolite. Moreover, the elemental compositions of HA, SFHA, and the zeolite are shown in Table 2. Results indicated the principal elements in HA and SFHA were C with 35.42% and 43.13%, and O with 20.32% and 46.43%, respectively. The Na content of SFHA considerably dropped, decreasing from 27.68% to 1.78% due to its reduction by washing. Other minor elements were involved in HA materials, such as Al, Si, K, P, S, and Cl. Meanwhile, Mg and Fe were not detected. Parameters are found within published ranges in the literature [34,35]. Information on the elemental composition of HA is not particularly conclusive due to their complexity and a wide range of other factors [36]. In addition, the main elements in the zeolite were Si (40.93%), C (31.99%), O (14.13%), and Al (7.46%), and minor contents of Na (0.25%), K (1.80%), Mg (1.12%), Ca (1.45%), and Fe (0.87%). Furthermore, Cl, P, and S were not detected. Similar chemical compositions for this kind of material have been reported by other researchers [37,38,39].

#### 2.1.4. ζ-Potential

In colloidal systems, the ζ-potential is frequently employed to track the behavior of particles suspended in a liquid. Additionally, the magnitude of the ζ-potential indicates the strength of the electrostatic attraction or repulsion between particles; therefore, it may be used to describe the surface of charged particles [40]. Figure 4 shows the relationship between pH and ζ-potential of HA, SFHA, and the zeolite. The three adsorbents generally show the same behavior: as the pH increases (from 2 to 11), the ζ-potential value becomes more negative. The pH values used for the determination of ζ-potential were 2, 5, and 7, each one according to the simulated compartment (crop, proventriculus, and intestine) in the in vitro digestive model. At pH 2, values of −46.44 mV and −30.75 mV were observed; at pH 5 of −50.16 mV and −49.2 mV; and at pH 7 of −54.46 mV and −40.93 mV for HA and SFHA, respectively. As a result, at all three pH values tested, HA was more negative, followed by SFHA and the zeolite. Our findings are consistent with those reported by Hamza et al [41] who state as the pH rises, the acidic functional groups of HA deprotonate, resulting in a more negative surface. Moreover, Deng and Bai [42] also found when pH exceeds 1.9, HA has a negative ζ-potential. Omar et al. (2014) and Coles and Yong [43] also reported ζ-potential values of HA of −20 mV at pH 3 and −44 mV at pH 10. Finally, Loosli et al. [44] reported strong negative charge with ζ-potential values ranging from −30.2 mV at pH 3 to −69.0 mV at pH 11. Even at low pH, HAs are negatively charged because of the dissociation of their acidic functional groups (mainly carboxylic and phenolic hydroxyl) [26,45]. Instead of aromatic components, variations in acid group conformation can be used to explain how the interactions are pH-dependent [32]. The first decrease (pH 3) of ζ-potential corresponds to the dissociation of carboxylic acid groups, while the second (pH 6) corresponds to the start of ionization of phenolic acid groups [46]. Our findings show the three adsorbents have a highly negative ζ-potential at the three simulated pH levels of the in vitro model, particularly at pH 7. As a result, the AFB_1_ molecule and the HA particle surface might be attracted by electrostatic forces since HAs are anionic polyelectrolytes that can interact with the positively charged AFB_1_ molecules [47].

#### 2.1.5. pHpzc

Understanding the surface charge of the particles is greatly aided by the pHpzc, which stands for the point of zero charge where the sum of positive and negative charges is equal. Figure 5 shows the pHpzc of HA, SFHA, and the zeolite. It has been hypothesized that the surface of the adsorbent will be positively charged if pH < pHpzc and negatively charged if pH ˃ pHpzc [47]. From the curves, the pHpzc for HA, SFHA, and the zeolite were 10.4, 2.2, and 8.8, respectively. According to Coles and Yong [43], the pHpzc of HA was less than 0.5. Moreover, Gjessing [48] reported pHpzc values for HA ranging from 1.2 to 1.8, which is very similar to the value of SFHA in our study. Furthermore, Giasuddin et al [49] reported the HA’s surface charge was negative over a pH range of 5 to 9.3. However, the presence of organic matter in the HA determines whether the pHpzc decreases or increases [43].

Interestingly, compared to the pHpzc of HA, which was 10.4, the SFHA reduced their pHpzc value to 2.2 after washing. This significant change in the pHpzc value could be attributed to a change in organic matter composition or to the purification process used to reduce the sodium salts in the HA. In this context, AFB_1_ adsorption is expected to be significant because SFHA has a negatively charged surface in all compartments simulated in the in vitro study. HA and the zeolite, on the other hand, have a positive surface charge on all three gastrointestinal sections, indicating electrostatic interactions do not govern AFB_1_ adsorption. It is well known that other mechanisms, such as weak electrostatic interactions and moderate electron donor–acceptor attraction, contribute to the adsorption of aflatoxins into inorganic binders [37].

### 2.2. In Vitro Digestive Model

Figure 6 depicts the percentage of AFB_1_ adsorption of the three tested sorbents. At the end of the in vitro model (the intestinal section), HA and SFHA adsorbents had 97.6% and 99.7% AFB_1_ uptake, respectively. Using the zeolite, a moderate sorption uptake of 81.5% was reached. In contrast, controls (without the addition of sorbent materials) show a marked lack of AFB_1_ adsorption (<3%). Following our findings, it was reported an in vitro model with 100 mg/mL of HA inclusion at 20 ng AFB_1_/g achieved 90.50% of AFB_1_ adsorption [50]. Moreover, Ye et al. [51] investigated the AFB_1_ adsorption capacity in the presence of sodium humate inclusions, various pH levels, interaction times, and AFB_1_ concentrations. The highest AFB_1_ adsorption percentages reported were 88.12% at pH 7 and 76.36% at pH 8. Furthermore, Vázquez-Durán et al. [27], using a dynamic in vitro model to assess the AFB_1_ adsorption capacity of a non-commercial zeolite at 5% (*w*/*w*) inclusion, reported an adsorption percentage of 75.5%, which is consistent with this research.

### 2.3. The Mechanism for AFB_1_ Adsorption onto HA

Because HA possesses highly hydrophobic surfaces and a wide variety of negatively charged functional groups [9], interactions between HA and AFB_1_ may involve different mechanisms. Tan [52] lists seven potential ways HA might bind to gaseous, liquid, and solid components: (i) physical forces, (ii) chemical forces, (iii) hydrogen bonds, (iv) hydrophobic interactions, (v) electrostatic interactions, (vi) coordination reactions, and (vii) ligand exchange. The most important interactions between HA and AFB_1_ are electrostatic interactions and hydrogen bonding. Nevertheless, other interactions could be considered, such as π–π stacking [53] and hydrophobic interactions (due to the many bond indices related to hydrophobic groups, including CH_2_, CH_3_, and C=C) [54] (Figure 7). Although this phenomenon is still not fully understood, aromatic structures and functional groups, such as OH and COOH, contributed to HA’s high AFB_1_ adsorption capacity [55]. Several techniques have been proposed to investigate the adsorption of different molecules onto HA. For example, physical modelling (Langmuir isotherm), kinetic modelling (Elovich kinetic model), surface complexation modelling, and Ligand and charge distribution model, among others [54]. However, to elucidate the nature of the molecular interactions between HA and AFB_1_, future theoretical simulations using density functional theory (DFT) may be considered [56].

Nevertheless, because in vitro tests cannot fully simulate the conditions of a bird’s digestive tract to determine the effectiveness of HA derived from vermicompost in reducing the harmful effects of AFB_1_, in vivo experiments must be conducted. Data on the effectiveness of HA extracted from vermicompost to reduce the impact of AFB_1_ in broilers are still meager. However, previous research has demonstrated the effectiveness of various humic substances as mycotoxins binders in in vivo trials. For instance, the addition of oxihumate (3.5 g/kg) was found to have a protective effect against liver, stomach, and heart damage in diets contaminated with AFB_1_, and a significative decrease in several serum, and hematological biochemical indicators linked to aflatoxin toxicity was also seen in broilers [57]. Adding HA to chicken feed in amounts ranging from 0.2% to 0.4% (*w*/*w*) improved feed efficiency, reduced liver and bursa damage, and improved serum biochemical profiles associated with aflatoxin toxicity [58]. Additionally, adding HA (0.3% *w*/*w*) decreased AFB_1_ residues in the liver and enhanced broiler antibody production against Newcastle disease [59].

## 3. Conclusions

In this study, two forms of HAs extracted from vermicompost were prepared, further characterized, and tested in AFB_1_ adsorption experiments using an in vitro poultry digestive model. Despite differences in the microstructure, elemental composition, surface functional groups, pHpzc, and ζ-potential, both adsorbents demonstrated significant AFB_1_ adsorption capacities when using an in vitro poultry digestive model. As a result, it can be concluded HA derived from vermicompost is highly effective in the adsorption of AFB_1_. However, more in vivo studies will enhance our comprehension of HA efficacy to reduce the toxic effects of AFB_1_ in poultry. There is now research being conducted in this area.

## 4. Materials and Methods

### 4.1. Humic Acids

As previously described [12], HAs were extracted and isolated from a vermicompost. HAs were extracted with a sodium hydroxide solution (1M NaOH) in compost:alkali ratio of 1:4, then stirred for 2 h. A Whatman grade 40 filter paper was used to filter the suspension after it had been at room temperature for 24 h. The supernatant was then separated by decantation after the filtrate had been centrifuged for 15 min at 3500× *g*. The HA-containing supernatant was acidified with 10% HCl and agitated continuously until pH 2 was achieved, allowing the HA to precipitate. Centrifugation at 3500× *g* for 15 min separated HA from FA. Finally, the precipitate (HA) was normalized with 1M NaOH until pH 10 was achieved, then oven-dried at 60 °C. A black powder was produced as a result. To produce the second adsorbent material, hereinafter referred to as sodium-free humic acids (SFHA), HAs were redispersed in deionized water and neutralized with 10% HCl until pH 7 was reached, then washed (centrifugation and redispersion) ten times to remove excess sodium and subsequently oven-dried at 60 °C.

### 4.2. Characterization of HA

#### 4.2.1. ATR-FTIR

Using a Fourier transform infrared spectrophotometer Frontier SP8000 (Perkin Elmer, Waltham, MA, USA) equipped with an attenuated total reflection (ATR) attachment (DuraSamplIR II, Smiths Detection, Warrington, UK), functional groups on the surface of the adsorbent materials were analyzed. Samples were deposited on the ATR diamond crystal, and spectra were collected in transmittance mode by combining 32 scans with a resolution of 4 cm^−1^ in the 4000–400 cm^−1^ region.

#### 4.2.2. Scanning Electron Microscopy (SEM)

Utilizing a scanning electron microscope (JSM-6010LA, Jeol Inc., MA, USA), the adsorbents’ size and morphology was examined. A thin gold coating was applied to the samples to improve electron conductivity and image quality. Using a 20 kV accelerating voltage, microscopy was carried out. Secondary electron imaging mode was used to capture the images at a 1000× magnification.

#### 4.2.3. Energy-Dispersive X-ray Spectroscopy (EDS)

Using an energy-dispersive X-ray spectrometer with an environmental scanning electron microscope (Phillips XL30, EDS-ESEM, Eindhoven, The Netherlands), the multi-element analysis was carried out. A high-performance X Trace micro-spot X-ray source was used to analyze each sample three times, and an attached XFlash^®^ 6/10 silicon drift detector was used to quantify the X-ray fluorescence spectrum it produced (Bruker Nano GmbH, Berlin, Germany).

#### 4.2.4. Zeta Potential (ζ-Potential)

The ZetaSizer Pro (Malvern Instruments, Worcestershire, UK) was used to determine zeta potential. The samples used for the measurements (20 mg dissolved in 10 mL distilled water) were adjusted to various pH levels using either HCl (0.1 M) or NaOH (0.1 M). To minimize the effects of viscosity and scattering, 100 µL of the aqueous phase were collected and diluted with 2 mL deionized water. Then, diluted samples were examined in a disposable capillary cell DTS1070 at room temperature with a 120-s equilibration time. Each measurement included eleven runs and three replicates of each sample to obtain a consistent result. The ZS Xplorer software was used to examine the results.

#### 4.2.5. Point of Zero Charge (pHpzc)

In accordance with the instructions of Zavala-Franco et al [37], the pHpzc was measured. Briefly, equal quantities of sorbents were introduced to a series of flasks filled with distilled water at various pH levels (2, 5, 7, 9, and 11). The pH of the supernatant was measured after samples were agitated at 250 rpm for 195 min. The pH was measured using a combination glass electrode (Conductronic PC-45, Puebla, Mexico). The plot of ∆pH against pH was used to determine the pHpzc.

### 4.3. In Vitro Adsorption Studies

#### Preparation of the AFB_1_-Contaminated Diet

Aflatoxin (100 µg AFB_1_/mL) was made as a main stock in dimethyl sulfoxide. After that, distilled water was used to dilute the AFB_1_ solution to 1 µg AFB_1_/mL. An experimental maize-soybean meal diet was made to closely match the nutritional needs of broiler chickens, as suggested by the National Research Council [60]. There were no antibiotics or anticoccidial drugs in the diet (Table 3; adapted from Solís-Cruz et al. [61]). To achieve 100 µg AFB_1_/kg, the diet was contaminated with 0.5 mL of the AFB_1_ solution. Thereafter, five samples were chosen at random, and the content of AFB_1_ was determined using the immunoaffinity column clean-up and liquid chromatography with fluorescence detection methodology. Levels of B-aflatoxins (AFB_1_ and AFB_2_), total fumonisins (FB_1_, FB_2_, and FB_3_), and ochratoxin A (OTA) were also determined in the diet using monoclonal antibody-based affinity columns (VICAM Science Technology, Watertown, MA, USA) and fluorescence detection. In general, the experimental diet had no detectable levels of B-aflatoxins and total fumonisins; assayed contents of these mycotoxins were below the detection limits of the immunoaffinity column techniques employed (<1 ng/g and <0.016 mg/kg, respectively). OTA was present at a level of 7 ng/g.

### 4.4. In Vitro Digestive Model

The AFB_1_ adsorptive capacity of the tested materials was assessed using a previously described in vitro gastrointestinal poultry model [62] with minor modifications. The assay was carried out with one control (zeolite) and two different treatments (HA and SFHA). This model simulated the physiological conditions of broiler chicken crop, proventriculus, and intestine. Every tube was incubated at 40 °C while being shaken at 19 rpm at an angle of 30°. For each compartment, the pH, enzymes, and time windows were adjusted. In the beginning, 5 g of the AFB_1_-contaminated feed and 10 mg of each adsorbent material were placed in polypropylene tubes (50 mL). Each tube received 10 mL of 0.03 M HCl to imitate the crop environment (pH reached values ~5.2). For 30 min, the tubes were incubated. A pH range of 1.4 to 2.0 was attained by adding 2.5 mL of 1.5 M HCl and 3000 U of pepsin (Merck KGaA, Darmstadt, Germany) per gram of feed in each tube following the incubation time. For a further 45 min, each tube was incubated. To simulate the third and final gastrointestinal compartment, 6.84 mg of 8×-pancreatin (Merck KGaA, Darmstadt, Germany) per gram of feed was added to 6.5 mL of 1.0 M NaHCO_3_. The pH in this region was reached, between 6.4 and 6.8. Tubes were incubated for an additional 120 min. The entire in vitro digesting process took 195 min. Afterward, the supernatant from all tubes was collected and stored at −20 °C for further analysis after centrifuging them all at 7000× *g* for 30 min. To determine the real AFB_1_ concentrations in each tube, controls (without the addition of adsorbent materials) were also prepared. The entire experiment was carried out in quintuplicate. The adsorption percentage of AFB_1_ for each tested material was calculated as follows:(1)$$Adsorption\;\left(\%\right)=\frac{\left(Ci-Cs\right)}{Ci}\times 100$$
where Ci is the concentration of AFB_1_ in the control (ng/mL); and Cs is the concentration of AFB_1_ in the supernatant of the treatments (ng/mL).

### 4.5. Aflatoxin Assay

Using monoclonal anti-body-based immunoaffinity columns (Afla-B, VICAM Science Technology, Watertown, MA, USA), AFB_1_ was removed from the supernatants and then utilized for ultraperformance liquid chromatography (UPLC). A modified version of the procedure that Hernández-Ramirez et al [63] previously described was employed. A UPLC system (Waters ACQUITY H-class) was used, equipped with a quaternary solvent manager and a reverse phase column (2.1 mm × 100 mm, 1.7 µm particles). A mobile phase of water, methanol, and acetonitrile (64:18:18) was used to elute AFB1. Samples (1 µL) from the anti-body-based immunoaffinity columns were injected and eluted with a flow rate of 700 µL/min. A fluorescence detector with settings of 365 nm excitation and 429 nm emission was used to detect the toxin. The AFB_1_ concentration was estimated using a calibration curve with a standard reference (AFB_1_, Merck KGaA, Darmstadt, Germany).

### 4.6. Method Validation

The performance of the clean-up procedure was tested by measuring the percentage of AFB_1_ recovery using the UPLC methodology, spiking four replicates of the experimental poultry diet with six different aflatoxin contents over the range of 8 to 250 ng AFB_1_/g, attaining a toxin recovery of 92% with a standard deviation of 3.4, standard error of 1.7, and a coefficient variation value of 4.4%. Moreover, the validation of the UPLC method was performed based on the guidelines for single-laboratory validation of analytical methods for trace-level concentrations of organic chemicals elaborated by the AOAC/FAO/IAEA/IUPAC [64]. The following parameters were evaluated: limit of detection (LOD), limit of quantification (LOQ), and linearity. For linearity, a six-point calibration curve was plotted at concentrations over the range of 10 to 1000 ng AFB_1_/L. In general, detection and quantification limits were found to be 2.0 and 6.7 ng AFB_1_/L, respectively. The linearity estimated with the coefficient of determination (R^2^) was 0.9984. These results indicated the methodology used was acceptable.

### 4.7. Experimental Design and Statistical Analysis

Data was subjected to one-way analysis of variance (one way-ANOVA) as a completely randomized design. Significant differences among the means were determined by the Tukey test. A value of *p* = 0.05 was used to detect significant differences between treatments.

## Figures and Tables

**Figure 1 toxins-15-00083-f001:**
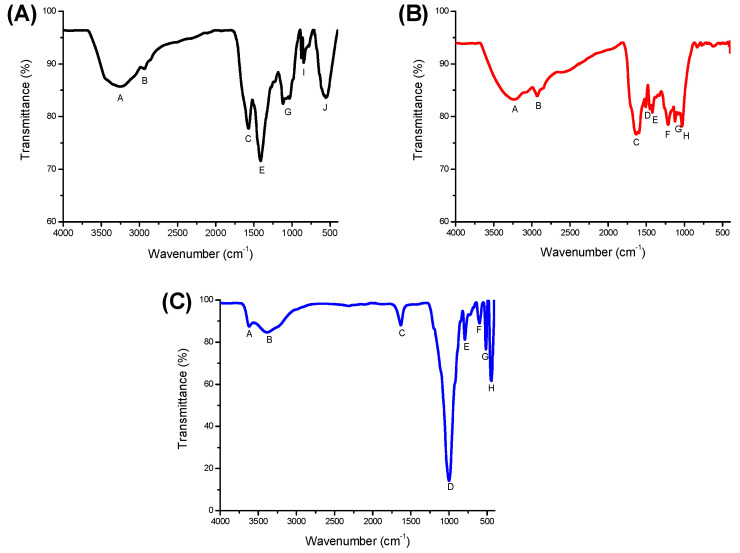
Comparative Fourier transform infrared spectra of: (**A**) humic acids, (**B**) sodium-free humic acids, and (**C**) the inorganic mycotoxin binder (zeolite).

**Figure 2 toxins-15-00083-f002:**
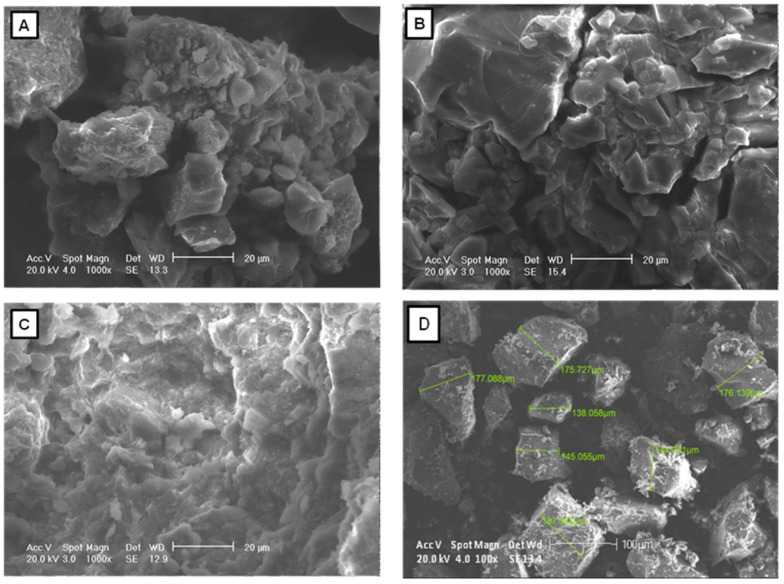
Structure of (**A**) humic acids, (**B**) sodium-free humic acids, and (**C**) the zeolite under SEM at ×1000. Measurements (**D**) of humic acids at ×100.

**Figure 3 toxins-15-00083-f003:**
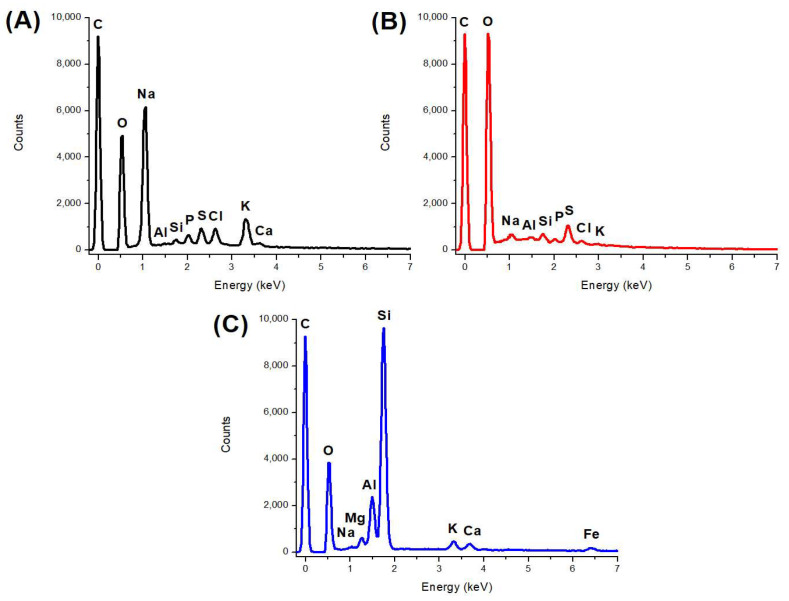
Representative energy-dispersive X-ray spectroscopy (EDS) spectra of (**A**) humic acids, (**B**) sodium-free humic acids, and (**C**) the zeolite.

**Figure 4 toxins-15-00083-f004:**
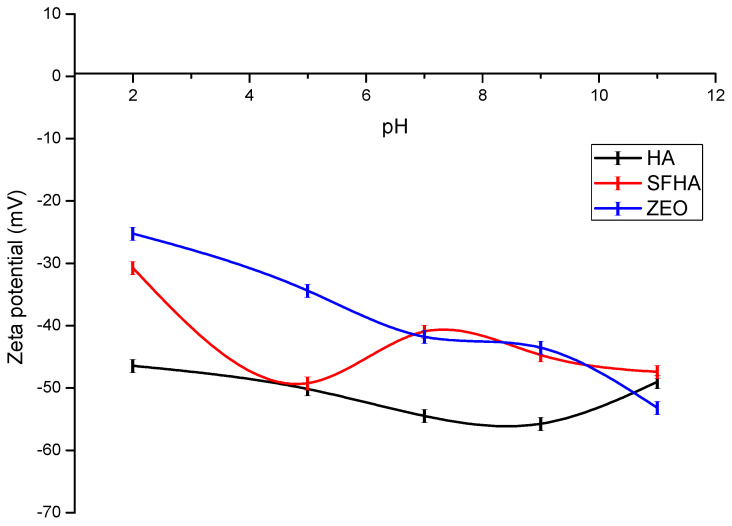
Relationship between zeta potential (ζ-potential) and pH of humic acids, sodium-free humic acids, and the zeolite. Mean of five replicates ± standard error.

**Figure 5 toxins-15-00083-f005:**
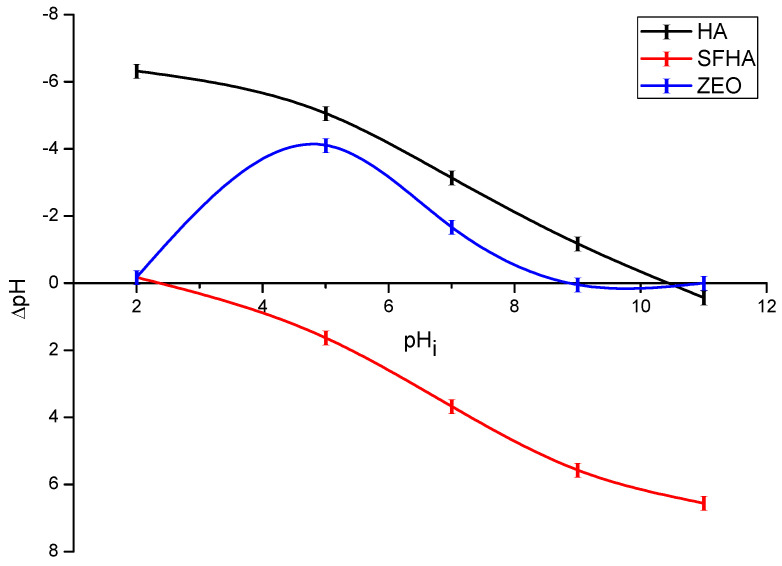
Point of zero charge (pHpzc) of humic acids, sodium-free humic acids, and the zeolite. Mean of five replicates ± standard error.

**Figure 6 toxins-15-00083-f006:**
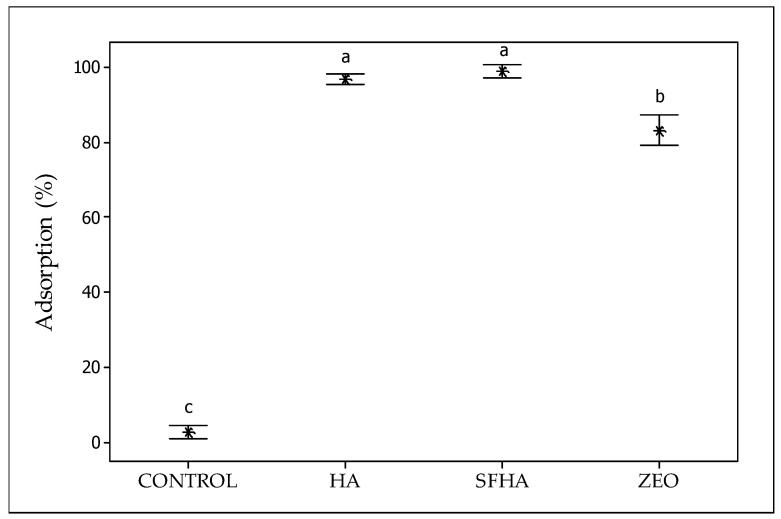
The adsorption capacity of humic acids (HA), sodium-free humic acids (SFHA), and the zeolite against AFB_1_ using an in vitro digestive poultry model. Mean values ± standard error. ^a–c^ Means with different letter are statistically different (Tukey *p* ≤ 0.05).

**Figure 7 toxins-15-00083-f007:**
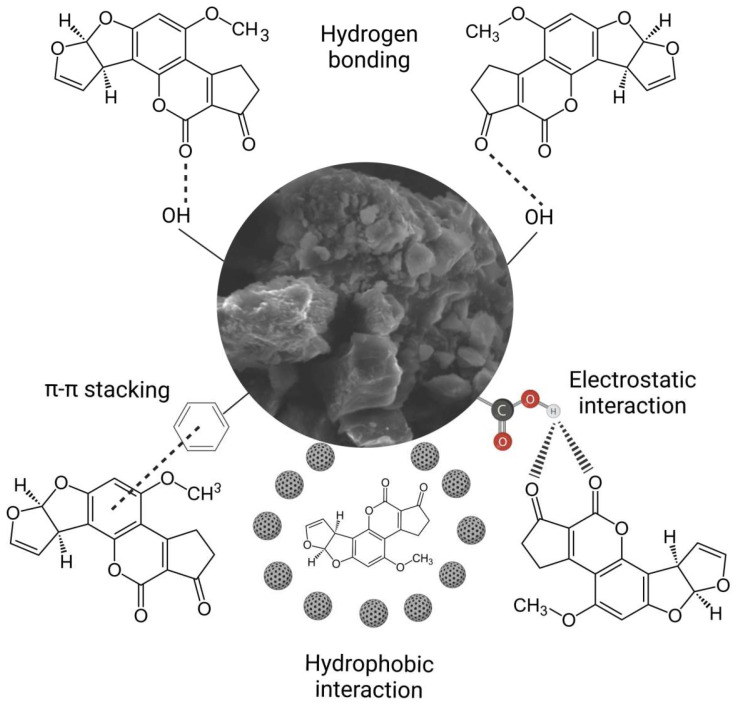
The hypothetical mechanism by which humic acids bind the AFB_1_ molecule.

**Table 1 toxins-15-00083-t001:** Band assignments of the vibrational frequencies in the humic acids and the inorganic mycotoxin binder (zeolite).

Band	Wavenumber (cm^−1^)		Associated Functional Group
	HA	SFHA	
A	3267	3238	O-H stretching vibrations and partially N-H stretch
B	2926	2927	Aliphatic CH_2_ and CH_3_
C	1571	1597	COO^−^, amides (NH_2_ of NH bonding)
D		1508	C=O of carboxylic groups and amide; N–H stretch; aromatic C=C
E	1412	1420	COO^−^ C=C stretch (aromatic ring), O–H and C–O of phenolic; C–N; C–H of CH_3_, CH_2_ and CH
F		1215	C–O and O–H stretch of COOH and phenol
G	1120	1123	C–O stretch of alcohols, carbonyl, esters and ethers, O-H of phenol and carbohydrates
H		1030	Aromatic ethers, C-O of carbohydrates; Si-O of silicate
I	846		Aromatic groups
J	620		Aliphatic –CH_2_
Zeolite			
A	3621	O-H stretching (Al^3+^-OH)	
B	3387	O-H stretching vibrations	
C	1630	H-O-H bending (water)	
D	1000	Si-O-Si antisymmetric stretch	
E	793	Si-O-Si stretching symmetric	
F	600	SiO_4_ and AlO_4_ tetrahedral	
G	515	Si-O bending vibration	
H	444	Si-O bending mode (-SiO_4_^−^)	

**Table 2 toxins-15-00083-t002:** The elemental composition (%) of humic acids (HA), sodium-free humic acids (SFHA) and the zeolite.

Element	Adsorbent			SEM	*p*-Value
	HA	SFHA	Zeolite		
C	35.42 ^b^	43.13 ^a^	31.99 ^c^	0.20	<0.0001
O	20.32 ^b^	46.43 ^a^	14.13 ^c^	0.61	<0.0001
Na	27.68 ^a^	1.78 ^b^	0.25 ^c^	0.11	<0.0001
Al	0.26 ^c^	0.78 ^b^	7.46 ^a^	0.17	<0.0001
Si	0.82 ^c^	1.60 ^b^	40.93 ^a^	0.08	<0.0001
K	6.66 ^a^	0.17 ^c^	1.80 ^b^	0.04	0.03
P	1.53 ^a^	0.59 ^b^	ND	0.24	0.02
S	3.00 ^b^	4.80 ^a^	ND	0.21	0.004
Cl	3.77 ^a^	0.71 ^b^	ND	^-^	^-^
Mg	ND	ND	1.12 a	^-^	^-^
Ca	0.54 ^b^	ND	1.45 ^a^	^-^	0.02
Fe	ND	ND	0.87	-	-

^abc^ Means with non-matching superscripts within rows indicates a significant difference at *p* < 0.05. ND = Not detected.

**Table 3 toxins-15-00083-t003:** Ingredient composition of the experimental poultry diet.

Ingredient	%
Maize	55.07
Soybean meal	36.94
Vegetable oil	3.32
Dicalcium phosphate	1.58
Calcium phosphate	1.44
Salt	0.35
DL-Methionine	0.25
Choline chloride 60%	0.20
L-Lysine HCL	0.10
Vitamin premix ^1^	0.30
Mineral premix ^2^	0.30
Antioxidant ^3^	0.15
Protein	19.5%
Metabolizable energy	13 MJ/kg

^1^ Vitamin premix supplied the following per kg: vitamin A, 2000 IU; vitamin D3, 600, IU; vitamin E, 7.5 IU; vitamin K3, 9 mg; thiamine, 3 mg; riboflavin, 8 mg, pantothenic acid, 18 mg; niacin, 60 mg; pyridoxine, 5 mg; folic acid, 2 mg; biotin, 0.2 mg; cyanocobalamin, 16 mg; and ascorbic acid, 200 mg. ^2^ Mineral premix supplied the following per kg; manganese, 120 mg; zinc, 100 mg; iron, 120 mg; copper, 10–15 mg; iodine, 0.7 mg; selenium, 0.4 mg; and cobalt, 0.2 mg. ^3^ Ethoxyquin.

## Data Availability

The datasets generated for this study are available on request to the corresponding author.

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
