# Peer review of "Humic Acids Preparation, Characterization, and Their Potential Adsorption Capacity for Aflatoxin B1 in an In Vitro Poultry Digestive Model"

_toxins, 2023, doi:10.3390/toxins15020083_

Round 1
Reviewer 1 Report (New Reviewer)
In this study, two forms of HA, natural HA and SFHA, were prepared and characterized, and their adsorbent capacity against AFB1 was evaluated. The study designed well and obtained interesting results. Some comments for the manuscript need to be addressed before it can be accepted
1. In the section of “Introduction” (lines 44-46), Except for the detoxification means of AFB1, other ways to detoxify, such as animal function regulator (curcumin, grapeseed and sea buckthorn meal), were not introduced or discussed. It is necessary to briefly introduce the detoxify means in the Introduction section, and cite related studies as the following reports herein, please.
Wang, Y.; Liu, F.; Zhou, X.; Liu, M.; Zang, H.; Liu, X.; Shan, A.; Feng, X. Alleviation of Oral Exposure to Aflatoxin B1-Induced Renal Dysfunction, Oxidative Stress, and Cell Apoptosis in Mice Kidney by Curcumin. Antioxidants 2022, 11, 1082. https://doi.org/10.3390/antiox11061082
Popescu, R.G.; Avramescu, S.; Marin, D.E.; Țăranu, I.; Georgescu, S.E.; Dinischiotu, A. The Reduction of the Combined Effects of Aflatoxin and Ochratoxin A in Piglet Livers and Kidneys by Dietary Antioxidants. Toxins 2021, 13, 648. https://doi.org/10.3390/toxins13090648
2. Line 81: “(C) 1571 - 1957 cm-1 and with”, the word “and” should be “associated”
3. Lines 197: “as the pH increases (from 2 to 11), the ζ-potential value becomes more negative”. The ζ-potential value of SFHA in Figure 4 did not show this trend. The ζ-potential value of SFHA tends to increase at pH7 and pH9.
4. Line 202: “AHP”, I think it should be “SFHA”.
5. In the figure captions of Fig. 4 and Fig. 5, delete “(A)”, “B”, and “C”, due to no corresponding letters in the two figures.
6. Line 370: “Aflatoxin (100 g AFB1/mL)”, the concentration is impossible.
7. Page 12, Table 3, The composition of the formula is not 100%, but 99.57%. In the note of the table, “vitamin A, 20,000,000 IU; vitamin D3, 6,000,000 IU; vitamin E, 75,000 IU;” the dosage of three vitamins is too high. In practical application, it is toxic to feed animals with this dosage in feed.
8. Line 445: “α = 0.05”, α should be p.
Author Response
Dear Reviewer, thank you very much for the time you have spent on reviewing our manuscript. Your comments are very valuable and helpful for revising our paper and guiding our research. We have studied those comments carefully and have made corrections, which we hope to meet with the approval. Revised portions in the new version were included and are highlighted in yellow in the reviewed manuscript. The following is our point-by-point response.
- In the section of “Introduction” (lines 44-46), Except for the detoxification means of AFB1, other ways to detoxify, such as animal function regulator (curcumin, grapeseed and sea buckthorn meal), were not introduced or discussed. It is necessary to briefly introduce the detoxify means in the Introduction section, and cite related studies as the following reports herein, please.
Wang, Y.; Liu, F.; Zhou, X.; Liu, M.; Zang, H.; Liu, X.; Shan, A.; Feng, X. Alleviation of Oral Exposure to Aflatoxin B1-Induced Renal Dysfunction, Oxidative Stress, and Cell Apoptosis in Mice Kidney by Curcumin. Antioxidants 2022, 11, 1082. https://doi.org/10.3390/antiox11061082
Popescu, R.G.; Avramescu, S.; Marin, D.E.; Țăranu, I.; Georgescu, S.E.; Dinischiotu, A. The Reduction of the Combined Effects of Aflatoxin and Ochratoxin A in Piglet Livers and Kidneys by Dietary Antioxidants. Toxins 2021, 13, 648. https://doi.org/10.3390/toxins13090648
A: Following your recommendations, additional information was added about the other methods of detoxifying food contaminated with mycotoxins, including the two above references.
- Line 81: “(C) 1571 - 1957 cm-1and with”, the word “and” should be “associated”
A: Thank you very much for the observation, the change has been made.
- Lines 197: “as the pH increases (from 2 to 11), the ζ-potential value becomes more negative”. The ζ-potential value of SFHA in Figure 4 did not show this trend. The ζ-potential value of SFHA tends to increase at pH7 and pH9.
A: The three adsorbents generally show the same behavior: as the pH increases (from 2 to 11), the ζ-potential value becomes more negative. The pH values used for the determination of ζ-potential were 2, 5, and 7, each one according to the simulated compartment (crop, proventriculus, and intestine) in the in vitro digestive model. At pH 2, values of -46.44 mV and -30.75 mV were observed; at pH 5 of -50.16 mV and -49.2 mV; and at pH 7 of -54.46 mV and -40.93 mV for HA and SFHA, respectively.
- Line 202: “AHP”, I think it should be “SFHA”.
A: Thank you very much for the observation, the change has been made.
- In the figure captions of Fig. 4 and Fig. 5, delete “(A)”, “B”, and “C”, due to no corresponding letters in the two figures.
A: Thank you very much for the observation, the change has been made.
- Line 370: “Aflatoxin (100 g AFB1/mL)”, the concentration is impossible.
A: Thank you very much for the observation, the change has been made.
- Page 12, Table 3, The composition of the formula is not 100%, but 99.57%. In the note of the table, “vitamin A, 20,000,000 IU; vitamin D3, 6,000,000 IU; vitamin E, 75,000 IU;” the dosage of three vitamins is too high. In practical application, it is toxic to feed animals with this dosage in feed.
A: Thank you very much for the observation, the change has been made. In brief, three ceros were removed from each vitamin.
- Line 445: “α = 0.05”, α should be p.
A: Thank you very much for the observation, the change has been made.
Reviewer 2 Report (New Reviewer)
The authors present a very good and topical work where they use vermicompost for the preparation of humic acid (HA) and the adsorption of aflatoxin B1 (AFB1) was investigated. They evaluated two forms of HA, natural HA and sodium-free HA (SFHA). They do something very novel and that is to use a non-commercial zeolitic material as adsorbent. They use appropriate techniques for the characterisation of the systems attenuated total reflectance-fourier transform infrared spectroscopy (ATR-FTIR), energy dispersive X-ray spectroscopy (EDS), zeta potential (ζ-potential), scanning electron microscopy (SEM) and point of zero charge (pHpzc). The
adsorption capacity of the materials when The adsorption capacity of the materials when added to a diet contaminated with AFB1 (100 µg/kg) was evaluated using an in vitro model simulating the in vitro model simulating the digestive tract of chickens. Characterisation results revealed that the primary functional groups in HA and SFHA were carboxyl and phenol. Furthermore, the adsorbents have a highly negative ζ- Furthermore, the adsorbents have a highly negative ζ-potential at all three simulated pH values. Therefore, it seems that the main factors for the adsorption of AFB1 are electrostatic interactions and hydrogen bonds. Furthermore, the bioavailability of AFB1 in the intestinal section decreased drastically when added to the diet (0.2%, w/w). The highest percentages of AFB1 adsorption using HA and SFHA were 97.6% and 99.7%, respectively. Zeolitic material had considerable adsorption (81.5%). From these results, it can be concluded that HA and SFHA from vermicompost could be used as potential adsorbents to remove AFB1 from contaminated feed.
The work is good and I recommend its publication in the present form.
Author Response
Thank you very much for the time you have spent on reviewing our manuscript. Your comments are very valuable and helpful for revising our paper and guiding our research.
Reviewer 3 Report (New Reviewer)
The Manuscript "Humic acids preparation, characterization, and their potential adsorption capacity for Aflatoxin B1 in an in vitro poultry digestive model" deals with a study where the capacity of "humic acids" to remove AFB1 were tested. Even if the quality of the presentation (language use and style) is high, I regret to suggest rejection of the manuscript due to one main concern. Humification in soil/during composting has been largely debated in the last decades, and the traditional view (degradation of organic matter followed by repolymerization) is now no longer acceptable. Please see:
Doi: 10.1038/nature16069
A consolidated assessment of published evidence reveals that secondary synthesis of ‘humic substances’ facilitated by minerals or enzymes (humification) has not been shown to be relevant in natural systems. In addition, the development of the extraction method in alkali preceded theory, tempting scientists to develop explanations for the synthesis of materials resembling operationally extracted ‘humic substances’, rather than to develop an understanding of the nature of all organic matter in soil or during composting. Over time, this attempt to mechanistically explain the formation of operationally defined ‘humic substances’ also led to their definition as synthesis products without the link to the alkaline extraction.
In the light of these evidence, I have to suggest the rejection of this manuscript that is fully based on the concepts of "humification" and "humic substances".
Author Response
Thank you very much for the time you have spent on reviewing our manuscript. Your comments are very valuable and helpful for revising our paper and guiding our research.
Response: Dear reviewer, we would like to highlight two key factors that stand in our favour that justify the use of humic substances in our research.
- In the article of Lehmann and Kleber (2015) entitled “The contentious nature of soil organic matter” a conceptual problem about the definition of humic substances is approached aided to improve the modelling of soil organic carbon. It is concluded that “soil organic matter is a continuum of progressively decomposing organic compounds” in which the formation of “humic substances (HS)” was argued. A theoretical “soil continuum model” (Fig. 2), is introduced.
However, the suggested soil continuum model by Lehmann and Kleber (2015) has been rejected by the academic community that studies HS in different fields (for instance, geochemistry and soil science). Please see some examples:
https://doi.org/10.1016/j.apsoil.2020.103655
https://doi.org/10.1016/j.apsoil.2020.103655
In the last report (De Novili et al., 2020) it has been stated “This review shows that a vast volume of interdisciplinary scientific evidence supports the formation of relevant non-pre-existing complex molecules exhibiting various types of structures. These molecules form during degradation and decay of biological cell components. The spontaneous abiotic and enzymatically catalysed reactions of components of organic residues and of their oxidative decomposition products suggested by state-of-the-art studies are indeed those proposed by most of the classical descriptions of humification. The review also highlights the chemically active role of pedofauna, explaining why the apparently harsh conditions of alkaline extraction of HS cannot be considered un-natural.”
- Our study did not deal with the conceptual problems of formation of “soil organic matter”. We did not used soil but rather a worm compost to extract the HS; and we did not model the organic carbon in the worm compost either, and because the worm compost was produced in a period of 3-5 months, with did not deal with continuum processes of organic matter transformation that normally occurs in water, soil and the environment.
We also recognize that the concept of HS is controversial, but to date, the concept is still used and accepted by most of the scientific community that studies humification and HS in different fields. There were thousands of publications before, and there has been hundreds of publications after Lehmann and Kleber (2015) report, where the term HS is used. Likewise, alkaline extraction is universally accepted as the most effective way to separate HS of different origin. Please see some examples:
https://DOI.org/10.1007/978-3-030-17891-8_7
https://doi.org/10.1007/978-3-030-17891-8_7
https://doi.org/10.1016/j.agee.2022.107928
Using Google Schoolar and the key words “Humic Acid” during 2019 and 2023, you can see that there are 28,100 scientific references
https://scholar.google.com/scholar?hl=en&as_sdt=0%2C4&q=humic+acid&oq=humic
In the study of Savarese et al. (2022) it was stated “it has been shown that an advanced molecular understanding of the soil Humeome, which is the ensemble of organic molecules in soil humus, can be achieved by a chemical fractionation sequence, named Humeomics (Nebbioso and Piccolo, 2011, 2012; Nebbioso et al., 2014a, 2014b, 2015). This extraction technique was developed following the novel concept of soil humus that, rather than being constituted by macropolymers as traditionally believed (Piccolo, 2002), is now regarded as a supramolecular association of small heterogeneous molecules held together by weak linkages such as van der Waals, π-π, hydrogen, and metal bridged electrostatic bonds (Piccolo, 2002; Lehmann and Kleber, 2015; Wells, 2019; Piccolo et al., 2018b).”
Humic matter has undoubtedly been extensively researched in geochemistry and soil science. Our knowledge of the inner molecular building principles has increased thanks to contemporary analytical tools. Here, HA was subjected to sophisticate solid-state Nuclear Magnetic Resonance (NMR) and high performance size exclusion chromatography (HPSEC), Infrared spectroscopies (IRS) and pyrolysis-mass spectrometry as well as metabolomics methods. The conformational behavior of HS has only recently been explained by the supramolecular nature of HS, based on the results of first, low-pressure size-exclusion chromatography.
In two previous studies of our group, the concentration of functional groups, elemental analysis, type of crystals, percentage of aromaticity, the estimated chemical properties and the flat structures of HS from vermi compost were previously published.
http://dx.doi:10.3390/ani9121101
http://dx.doi.org/10.1590/1806-9061-2021-1450
For these reasons, we respectfully encourage you to reconsider your position about HS; other than that, we highly appreciate your previous comment: “Even if the quality of the presentation (language use and style) is high”.
Round 2
Reviewer 1 Report (New Reviewer)
All my comments have been addressed in the revised manuscript, I have no other comments.
Reviewer 3 Report (New Reviewer)
The authors have deeply explained their point of view on the issue raised in the first revision. Even if I still remain on my positions about HS, the effort of the authors in explaining their position is apprecciated. In addition, they brought lot of papers supporting their positions. I cannot doing anything different from suggesting acceptance of this paper.
This manuscript is a resubmission of an earlier submission. The following is a list of the peer review reports and author responses from that submission.
Round 1
Reviewer 1 Report
The manuscript entitled “Humic acids preparation, characterization, and their potential adsorption capacity for Aflatoxin B1 in an in vitro poultry digestive model” present the results of the humic acid (HA) preparation and the adsorption of aflatoxin B1 (AFB1).
Authors should correct manuscript according to the suggestion
Minor issues:
Introduction:
what was the aim of the study? please clearly (in a few sentences) specify the purpose at the end of the introduction?
Line 28-20: Authors should add information on the toxigenic fungi, not only Aspergillus sp. produced mycotoxins.
Line 41: please cite literature to approve statement “The detoxification of mycotoxin-contaminated grains can be accomplished using a variety of procedures, including physical removal, chemical conversion to less toxic products, enzymatic detoxification, and microbial degradation, among others”
References:
correct the references in accordance with the requirements of the journal, e.g. in 3, 6, 13 and 15 the full names of the journals should be given
Reviewer 2 Report
The manuscript evaluates the potential of a newly prepared and characterized humic acid for the mitigation of aflatoxin B1 (AFB1) in animal feed. The preparation and characterization of humic acid described in this paper is not novel as there are several similar reports available in the scientific literature on the use of humic acid for mycotoxin removal. Additionally, the authors have not provided substantial evidence to prove that the humic acid is efficient for AFB1 removal. For instance, the HPLC method that was used for AFB1 analysis was not validated, and there is no results of the background AFB1 levels of the feed before it was spiked. Furthermore, no where in the manuscript was the formula used to obtain the percentage adsorption mentioned. Thus, the results are not reliable.
I therefore recommend a rejection.
Reviewer 3 Report
The paper contains a quite exhaustive characterization of three substrates for AFB1 adsorption: HA, SFHA and Zeolite. Sadly, the date and logic for the adsorption process and adducts are superficial and mostly limited to the partition constant. Figure 6 is useless.
The paper is well exposed and structured. The language is clear and correct with some minor glitches or small errors; I attach the paper where I have underlined the points where I would modify the text.
Also, I have some questions I wish the authors would clarify:
- Why do you ascribe bands D and E of zeolite to " symmetric and an-
tisymmetric stretch vibrations of the Si-O-Si bonds", why not say of "O-Si-O"? Is there any phonon calculation behind this attribution?
- "which reduces intramolecular repulsion and encourages a more
expanded and scattered macrostructure.": this seems somewhat counterintuitive, why would a reduction of repulsion bring to expanded macrostructure?
- "Results indicated that the principal element in HA and SFHA
is O, with 64.84% and 89.05%, respectively": I do not understand, where is carbon? it is not even mentioned in the analysis. Also, why does the plot of figure 3,a start at 0.5 keV contrary to those of 3,b and 3,c?
